# GROUNDING LANGUAGE TO AUTONOMOUSLY-ACQUIRED SKILLS VIA GOAL GENERATION

**Ahmed Akakzia**\*
Sorbonne Université
ahmed.akakzia@isir.upmc.fr

**Cédric Colas**\*
Inria
cedric.colas@inria.fr

**Pierre-Yves Oudeyer**
Inria

**Mohamed Chetouani**
Sorbonne Université

**Olivier Sigaud**
Sorbonne Université

## ABSTRACT

We are interested in the autonomous acquisition of repertoires of skills. Language-conditioned reinforcement learning (LC-RL) approaches are great tools in this quest, as they allow to express abstract goals as sets of constraints on the states. However, most LC-RL agents are not autonomous and cannot learn without external instructions and feedback. Besides, their direct language condition cannot account for the goal-directed behavior of pre-verbal infants and strongly limits the expression of behavioral diversity for a given language input. To resolve these issues, we propose a new conceptual approach to language-conditioned RL: the Language-Goal-Behavior architecture (LGB). LGB decouples skill learning and language grounding via an intermediate semantic representation of the world. To showcase the properties of LGB, we present a specific implementation called DECSTR. DECSTR is an intrinsically motivated learning agent endowed with an innate semantic representation describing spatial relations between physical objects. In a first stage (G→B), it freely explores its environment and targets self-generated semantic configurations. In a second stage (L→G), it trains a language-conditioned goal generator to generate semantic goals that match the constraints expressed in language-based inputs. We showcase the additional properties of LGB w.r.t. both an end-to-end LC-RL approach and a similar approach leveraging non-semantic, continuous intermediate representations. Intermediate semantic representations help satisfy language commands in a diversity of ways, enable strategy switching after a failure and facilitate language grounding.

## 1 INTRODUCTION

Developmental psychology investigates the interactions between learning and developmental processes that support the slow but extraordinary transition from the behavior of infants to the sophisticated intelligence of human adults (Piaget, 1977; Smith & Gasser, 2005). Inspired by this line of thought, the central endeavour of developmental robotics consists in shaping a set of machine learning processes able to generate a similar growth of capabilities in robots (Weng et al., 2001; Lungarella et al., 2003). In this broad context, we are more specifically interested in designing learning agents able to: 1) explore open-ended environments and grow repertoires of skills in a self-supervised way and 2) learn from a tutor via language commands.

The design of intrinsically motivated agents marked a major step towards these goals. The Intrinsically Motivated Goal Exploration Processes family (IMGEPs), for example, describes embodied agents that interact with their environment at the sensorimotor level and are endowed with the ability to represent and set their own goals, rewarding themselves over completion (Forestier et al., 2017). Recently, goal-conditioned reinforcement learning (GC-RL) appeared like a viable way to implement IMGEPs and target the open-ended and self-supervised acquisition of diverse skills.

---

\*Equal contribution.

Goal-conditioned RL approaches train goal-conditioned policies to target multiple goals (Kaelbling, 1993; Schaul et al., 2015). While most GC-RL approaches express goals as target features (e.g. target block positions (Andrychowicz et al., 2017), agent positions in a maze (Schaul et al., 2015) or target images (Nair et al., 2018)), recent approaches started to use language to express goals, as language can express sets of constraints on the state space (e.g. *open the red door*) in a more abstract and interpretable way (Luketina et al., 2019).

However, most GC-RL approaches – and language-based ones (LC-RL) in particular – are not intrinsically motivated and receive external instructions and rewards. The IMAGINE approach is one of the rare examples of intrinsically motivated LC-RL approaches (Colas et al., 2020). In any case, the language condition suffers from three drawbacks. 1) It couples skill learning and language grounding. Thus, it cannot account for goal-directed behaviors in pre-verbal infants (Mandler, 1999). 2) Direct conditioning limits the behavioral diversity associated to language input: a single instruction leads to a low diversity of behaviors only resulting from the stochasticity of the policy or the environment. 3) This lack of behavioral diversity prevents agents from switching strategy after a failure.

To circumvent these three limitations, one can decouple skill learning and language grounding via an intermediate innate semantic representation. On one hand, agents can learn skills by targeting configurations from the semantic representation space. On the other hand, they can learn to generate valid semantic configurations matching the constraints expressed by language instructions. This generation can be the backbone of behavioral diversity: a given sentence might correspond to a whole set of matching configurations. This is what we propose in this work.

**Contributions.** We propose a novel conceptual RL architecture, named LGB for Language-Goal-Behavior and pictured in Figure 1 (right). This LGB architecture enables an agent to decouple the intrinsically motivated acquisition of a repertoire of skills (Goals → Behavior) from language grounding (Language → Goals), via the use of semantic goal representation. To our knowledge, the LGB architecture is the only one to combine the following four features:

- It is intrinsically motivated: it selects its own (semantic) goals and generates its own rewards,
- It decouples skill learning from language grounding, accounting for infants learning,
- It can exhibit a diversity of behaviors for any given instruction,
- It can switch strategy in case of failures.

Besides, we introduce an instance of LGB, named DECSTR for **DE**ep sets and **C**urriculum with **S**eman**T**ic goal **R**epresentations. Using DECSTR, we showcase the advantages of the conceptual decoupling idea. In the *skill learning* phase, the DECSTR agent evolves in a manipulation environment and leverages semantic representations based on predicates describing spatial relations between physical objects. These predicates are known to be used by infants from a very young age (Mandler, 2012). DECSTR autonomously learns to discover and master all reachable configurations in its semantic representation space. In the *language grounding* phase, we train a Conditional Variational Auto-Encoder (C-VAE) to generate semantic goals from language instructions. Finally, we can evaluate the agent in an *instruction-following* phase by composing the two first phases. The experimental section investigates three questions: how does DECSTR perform in the three phases? How does it compare to end-to-end LC-RL approaches? Do we need intermediate representations to be semantic? Code and videos can be found at https://sites.google.com/view/decstr/.

## 2 RELATED WORK

**Standard language-conditioned RL.** Most approaches from the LC-RL literature define *instruction following* agents that receive external instructions and rewards (Hermann et al., 2017; Chan et al., 2019; Bahdanau et al., 2018; Cideron et al., 2019; Jiang et al., 2019; Fu et al., 2019), except the IMAGINE approach which introduced intrinsically motivated agents able to set their own goals and to imagine new ones (Colas et al., 2020). In both cases, the language-condition prevents the decoupling of language acquisition and skill learning, true behavioral diversity and efficient *strategy switching* behaviors. Our approach is different, as we can decouple language acquisition from skill learning. The language-conditioned goal generation allows behavioral diversity and strategy switching behaviors.

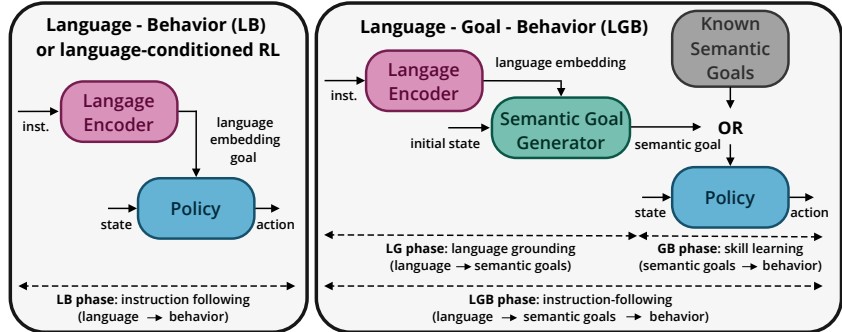

Figure 1: A standard language-conditioned RL architecture (left) and our proposed LGB architecture (right).

**Goal-conditioned RL with target coordinates for block manipulation.** Our proposed implementation of LGB, called DECSTR, evolves in a block manipulation domain. Stacking blocks is one of the earliest benchmarks in artificial intelligence (e.g. Sussman (1973); Tate et al. (1975)) and has led to many simulation and robotics studies (Deisenroth et al., 2011; Xu et al., 2018; Colas et al., 2019a). Recently, Lanier et al. (2019) and Li et al. (2019) demonstrated impressive results by stacking up to 4 and 6 blocks respectively. However, these approaches are not intrinsically motivated, involve hand-defined curriculum strategies and express goals as specific target block positions. In contrast, the DECSTR agent is intrinsically motivated, builds its own curriculum and uses semantic goal representations (symbolic or language-based) based on spatial relations between blocks.

**Decoupling language acquisition and skill learning.** Several works investigate the use of semantic representations to associate meanings and skills (Alomari et al., 2017; Tellex et al., 2011; Kulick et al., 2013). While the two first use semantic representations as an intermediate layer between language and skills, the third one does not use language. While DECSTR acquires skills autonomously, previous approaches all use skills that are either manually generated (Alomari et al., 2017), hand-engineered (Tellex et al., 2011) or obtained via optimal control methods (Kulick et al., 2013). Closer to us, Lynch & Sermanet (2020) also decouple skill learning from language acquisition in a goal-conditioned imitation learning paradigm by mapping both language goals and images goals to a shared representation space. However, this approach is not intrinsically motivated as it relies on a dataset of human tele-operated strategies. The deterministic merging of representations also limits the emergence of behavioral diversity and efficient strategy-switching behaviors.

## 3 METHODS

This section presents our proposed Language-Goal-Behavior architecture (LGB) represented in Figure 1 (Section 3.1) and a particular instance of the LGB architecture called DECSTR. We first present the environment it is set in [3.2], then describe the implementations of the three modules composing any LGB architecture: 1) the semantic representation [3.3]; 2) the intrinsically motivated goal-conditioned algorithm [3.4] and 3) the language-conditioned goal generator [3.5]. We finally present how the three phases described in Figure 1 are evaluated [3.6].

### 3.1 THE LANGUAGE-GOAL-BEHAVIOR ARCHITECTURE

The LGB architecture is composed of three main modules. First, the *semantic representation* defines the behavioral and goal spaces of the agent. Second, the intrinsically motivated GC-RL algorithm is in charge of the skill learning phase. Third, the language-conditioned goal generator is in charge of the language grounding phase. Both phases can be combined in the instruction following phase. The three phases are respectively called G→B for Goal → Behavior, L→G for Language → Goal and L→G→B for Language → Goal → Behavior, see Figure 1 and Appendix A. Instances of the LGB architecture should demonstrate the four properties listed in the introduction: 1) be intrinsically motivated; 2) decouple skill learning and language grounding (by design); 3) favor behavioral diversity; 4) allow strategy switching. We argue that any LGB algorithm should fulfill the following constraints. For LGB to be intrinsically motivated (1), the algorithm needs to integrate the generation

and selection of semantic goals and to generate its own rewards. For LGB to demonstrate behavioral diversity and strategy switching (3, 4), the language-conditioned goal generator must efficiently model the distribution of semantic goals satisfying the constraints expressed by any language input.

## 3.2 ENVIRONMENT

The DECSTR agent evolves in the *Fetch Manipulate* environment: a robotic manipulation domain based on MUJOCO (Todorov et al., 2012) and derived from the Fetch tasks (Plappert et al., 2018), see Figure 2. Actions are 4-dimensional: 3D gripper velocities and grasping velocity. Observations include the Cartesian and angular positions and velocities of the gripper and the three blocks. Inspired by the framework of *Zone of Proximal Development* that describes how parents organize the learning environment of their children (Vygotsky, 1978), we let a social partner facilitate DECSTR's exploration by providing non-trivial initial configurations. After a first period of autonomous exploration, the social partner initializes

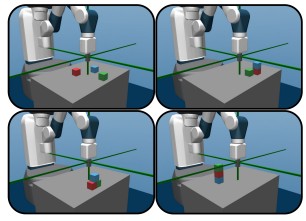

Figure 2: Example configurations. Top-right: (111000100).

the scene with stacks of 2 blocks $21\%$ of times, stacks of 3 blocks $9\%$ of times, and a block is initially put in the agent's gripper $50\%$ of times. This help is not provided during offline evaluations.

## 3.3 SEMANTIC REPRESENTATION

**Semantic predicates define the behavioral space.** Defining the list of semantic predicates is defining the dimensions of the behavioral space explored by the agent. It replaces the traditional definition of goal spaces and their associated reward functions. We believe it is for the best, as it does not require the engineer to fully predict all possible behaviors within that space, to know which behaviors can be achieved and which ones cannot, nor to define reward functions for each of them.

**Semantic predicates in DECSTR.** We assume the DECSTR agent to have access to innate semantic representations based on a list of predicates describing spatial relations between pairs of objects in the scene. We consider two of the spatial predicates infants demonstrate early in their development (Mandler, 2012): the *close* and the *above* binary predicates. These predicates are applied to all permutations of object pairs for the 3 objects we consider: 6 permutations for the *above* predicate and 3 combinations for the *close* predicate due to its order-invariance. A *semantic configuration* is the concatenation of the evaluations of these 9 predicates and represents spatial relations between objects in the scene. In the resulting semantic configuration space $\{0, 1\}^9$, the agent can reach 35 physically valid configurations, including stacks of 2 or 3 blocks and pyramids, see examples in Figure 2. The binary reward function directly derives from the semantic mapping: the agent rewards itself when its current configuration $c_p$ matches the goal configuration $c_p = g$. Appendix B provides formal definitions and properties of predicates and semantic configurations.

## 3.4 INTRINSICALLY MOTIVATED GOAL-CONDITIONED REINFORCEMENT LEARNING

This section describes the implementation of the intrinsically motivated goal-conditioned RL module in DECSTR. It is powered by the Soft-Actor Critic algorithm (SAC) (Haarnoja et al., 2018) that takes as input the current state, the current semantic configuration and the goal configuration, for both the critic and the policy. We use Hindsight Experience Replay (HER) to facilitate transfer between goals (Andrychowicz et al., 2017). DECSTR samples goals via its curriculum strategy, collects experience in the environment, then performs policy updates via SAC. This section describes two particularities of our RL implementation: the self-generated goal selection curriculum and the object-centered network architectures. Implementation details and hyperparameters can be found in Appendix C.

**Goal selection and curriculum learning.** The DECSTR agent can only select goals among the set of semantic configurations it already experienced. We use an automatic curriculum strategy

(Portelas et al., 2020) inspired from the CURIOUS algorithm (Colas et al., 2019a). The DECSTR agent tracks aggregated estimations of its *competence* (C) and *learning progress* (LP). Its selection of goals to target during data collection and goals to learn about during policy updates (via HER) is biased towards goals associated with high absolute LP and low C.

*Automatic bucket generation.* To facilitate robust estimation, LP is usually estimated on sets of goals with similar difficulty or similar dynamics (Forestier et al., 2017; Colas et al., 2019a). While previous works leveraged expert-defined *goal buckets*, we cluster goals based on their time of discovery, as the time of discovery is a good proxy for goal difficulty: easier goals are discovered earlier. Buckets are initially empty (no known configurations). When an episode ends in a new configuration, the $N_b = 5$ buckets are updated. Buckets are filled equally and the first buckets contain the configurations discovered earlier. Thus goals change buckets as new goals are discovered.

*Tracking competence, learning progress and sampling probabilities.* Regularly, the DECSTR agent evaluates itself on goal configurations sampled uniformly from the set of known ones. For each bucket, it tracks the recent history of past successes and failures when targeting the corresponding goals (last $W = 1800$ self-evaluations). C is estimated as the success rate over the most recent half of that history C = C$_{\text{recent}}$. LP is estimated as the difference between C$_{\text{recent}}$ and the one evaluated over the first half of the history (C$_{\text{earlier}}$). This is a crude estimation of the derivative of the C curve w.r.t. time: LP = C$_{\text{recent}}$ - C$_{\text{earlier}}$. The sampling probability P$_i$ for bucket $i$ is:

$$P_i = \frac{(1 - C_i) * |LP_i|}{\sum_j ((1 - C_j) * |LP_j|)}.$$

In addition to the usual LP bias (Colas et al., 2019a), this formula favors lower C when LP is similar. The absolute value ensures resampling buckets whose performance decreased (e.g. forgetting).

**Object-centered architecture.** Instead of fully-connected or recurrent networks, DECSTR uses for the policy and critic an *object-centered architecture* similar to the ones used in Colas et al. (2020); Karch et al. (2020), adapted from Deep-Sets (Zaheer et al., 2017). For each pair of objects, a shared network independently encodes the concatenation of body and objects features and current and target semantic configurations, see Appendix Figure 4. This shared network ensures efficient transfer of skills between pairs of objects. A second inductive bias leverages the symmetry of the behavior required to achieve *above*($o_i$, $o_j$) and *above*($o_j$, $o_i$). To ensure automatic transfer between the two, we present half of the features (e.g. those based on pairs ($o_i$, $o_j$) where $i < j$) with goals containing one side of the symmetry (all *above*($o_i$, $o_j$) for $i < j$) and the other half with the goals containing the other side (all *above*($o_j$, $o_i$) for $i < j$). As a result, the *above*($o_i$, $o_j$) predicates fall into the same slot of the shared network inputs as their symmetric counterparts *above*($o_j$, $o_i$), only with different permutations of object pairs. Goals are now of size 6: 3 *close* and 3 *above* predicates, corresponding to one side of the *above* symmetry. Skill transfer between symmetric predicates are automatically ensured. Appendix C.1 further describes these inductive biases and our modular architecture.

### 3.5 LANGUAGE-CONDITIONED GOAL GENERATION

The language-conditioned goal generation module (LGG) is a generative model of semantic representations conditioned by language inputs. It is trained to generate semantic configurations matching the agent's initial configuration and the description of a change in one object-pair relation.

A training dataset is collected via interactions between a DECSTR agent trained in phase G→B and a social partner. DECSTR generates semantic goals and pursues them. For each trajectory, the social partner provides a description $d$ of one change in objects relations from the initial configuration $c_i$ to the final one $c_f$. The set of possible descriptions contains 102 sentences, each describing, in a simplified language, a positive or negative shift for one of the 9 predicates (e.g. *get red above green*). This leads to a dataset $\mathcal{D}$ of 5000 triplets: ($c_i$, $d$, $c_f$). From this dataset, the LGG is learned using a conditional Variational Auto-Encoder (C-VAE) (Sohn et al., 2015). Inspired by the context-conditioned goal generator from Nair et al. (2019), we add an extra condition on language instruction to improve control on goal generation. The conditioning instruction is encoded by a recurrent network that is jointly trained with the VAE via a mixture of Kullback-Leibler and cross-entropy losses. Appendix C.2 provides the list of sentences and implementation details. By repeatedly sampling the LGG, a set of goals is built for any language input. This enables skill diversity and *strategy switching*: if the agent fails, it can sample another valid goal to fulfill the instruction, effectively

switching strategy. This also enables goal combination using logical functions of instructions: *and* is an intersection, *or* is an union and *not* is the complement within the known set of goals.

## 3.6 EVALUATION OF THE THREE LGB PHASES

**Skill learning phase** G→B**:** DECSTR explores its semantic representation space, discovers achievable configurations and learns to reach them. Goal-specific performance is evaluated offline across learning as the success rate (SR) over 20 repetitions for each goal. The global performance $\overline{\text{SR}}$ is measured across either the set of 35 goals or discovery-organized buckets of goals, see Section 3.4.

**Language grounding phase** L→G**:** DECSTR trains the LGG to generate goals matching constraints expressed via language inputs. From a given initial configuration and a given instruction, the LGG should generate all compatible final configurations (goals) and just these. This is the source of behavioral diversity and strategy switching behaviors. To evaluate LGG, we construct a synthetic, oracle dataset $\mathcal{O}$ of triplets $(c_i, d, \mathcal{C}_f(c_i, d))$, where $\mathcal{C}_f(c_i, d)$ is the set of all final configurations compatible with $(c_i, d)$. On average, $\mathcal{C}_f$ in $\mathcal{O}$ contains 16.7 configurations, while the training dataset $\mathcal{D}$ only contains 3.4 (20%). We are interested in two metrics: 1) The *Precision* is the probability that a goal sampled from the LGG belongs to $\mathcal{C}_f$ (true positive / all positive); 2) The *Recall* is percentage of elements from $\mathcal{C}_f$ that were found by sampling the LGG 100 times (true positive / all true). These metrics are computed on 5 different subsets of the oracle dataset, each calling for a different type of generalization (see full lists of instructions in Appendix C.2):

1. Pairs found in $\mathcal{D}$, except pairs removed to form the following test sets. This calls for the extrapolation of known initialization-effect pairs $(c_i, d)$ to new final configurations $c_f$ ($\mathcal{D}$ contains only 20% of $\mathcal{C}_f$ on average).
2. Pairs that were removed from $\mathcal{D}$, calling for a recombination of known effects $d$ on known $c_i$.
3. Pairs for which the $c_i$ was entirely removed from $\mathcal{D}$. This calls for the transfer of known effects $d$ on unknown $c_i$.
4. Pairs for which the $d$ was entirely removed from $\mathcal{D}$. This calls for generalization in the language space, to generalize unknown effects $d$ from related descriptions and transpose this to known $c_i$.
5. Pairs for which both the $c_i$ and the $d$ were entirely removed from $\mathcal{D}$. This calls for the generalizations 3 and 4 combined.

**Instruction following phase** L→G→B**:** DECSTR is instructed to modify an object relation by one of the 102 sentences. Conditioned on its current configuration and instruction, it samples a compatible goal from the LGG, then pursues it with its goal-conditioned policy. We consider three evaluation settings: 1) performing a single instruction; 2) performing a sequence of instructions without failure; 3) performing a logical combination of instructions. The *transition* setup measures the success rate of the agent when asked to perform the 102 instructions 5 times each, resetting the environment each time. In the *expression* setup, the agent is evaluated on 500 randomly generated logical functions of sentences, see the generation mechanism in Appendix C.2. In both setups, we evaluate the performance in 1-shot ($SR_1$) and 5-shot ($SR_5$) settings. In the 5-shot setting, the agent can perform *strategy switching*, to sample new goals when previous attempts failed (without reset). In the *sequence* setup, the agent must execute 20 sequences of random instructions without reset (5-shot). We also test behavioral diversity. We ask DECSTR to follow each of the 102 instructions 50 times each and report the number of different achieved configurations.

## 4 EXPERIMENTS

Our experimental section investigates three questions: [4.1]: How does DECSTR perform in the three phases? [4.2]: How does it compare to end-to-end language-conditioned approaches? [4.3]: Do we need intermediate representations to be semantic?

## 4.1 HOW DOES DECSTR PERFORM IN THE THREE PHASES?

This section presents the performance of the DECSTR agent in the skill learning, language grounding, and instruction following phases.

**Skill learning phase** G→B: Figure 3 shows that DECSTR successfully masters all reachable configurations in its semantic representation space. Figure 3a shows the evolution of $\overline{SR}$ computed per bucket. Buckets are learned in increasing order, which confirms that the time of discovery is a good proxy for difficulty. Figure 3b reports C, LP and sampling probabilities P computed online using self-evaluations for an example agent. The agent leverages these estimations to select its goals: first focusing on the easy goals from bucket 1, it moves on towards harder and harder buckets as easier ones are mastered (low LP, high C). Figure 3c presents the results of ablation studies. Each condition removes one component of DECSTR: 1) *Flat* replaces our object-centered modular architectures by flat ones; 2) *w/o Curr.* replaces our automatic curriculum strategy by a uniform goal selection; 3) *w/o Sym.* does not use the symmetry inductive bias; 4) In *w/o SP*, the social partner does not provide non-trivial initial configurations. In the *Expert buckets* condition, the curriculum strategy is applied on expert-defined buckets, see Appendix D.1. The full version of LGB performs on par with the *Expert buckets* oracle and outperforms significantly all its ablations. Appendix E.3 presents more examples of learning trajectories, and dissects the evolution of bucket compositions along training.

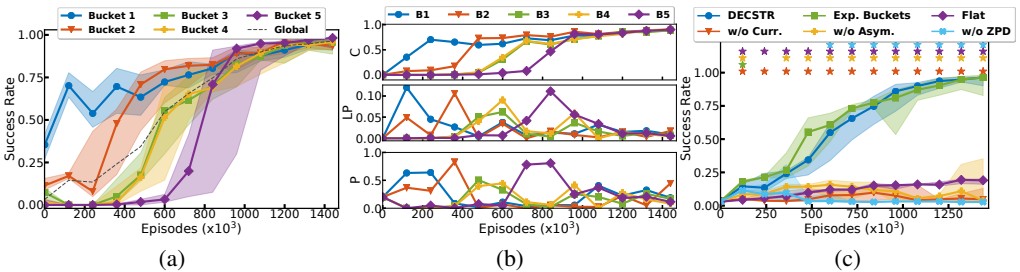

(a)             (b)             (c)

Figure 3: **Skill Learning**: (a) $\overline{SR}$ per bucket. (b): C, LP and P estimated by a DECSTR agent. (c): ablation study. Medians and interquartile ranges over 10 seeds for DECSTR and 5 seeds for others in (a) and (c). Stars indicate significant differences to DECSTR as reported by Welch's t-tests with $\alpha = 0.05$ (Colas et al., 2019b).

Table 1: L→G phase. Metrics are averaged over 10 seeds, stdev < 0.06 and 0.07 respectively.

| Metrics | Test 1 | Test 2 | Test 3 | Test 4 | Test 5 |
|---|---|---|---|---|---|
| Precision | 0.97 | 0.93 | 0.98 | 0.99 | 0.98 |
| Recall | 0.93 | 0.94 | 0.95 | 0.90 | 0.92 |

Table 2: L→G→B phase. Mean ± stdev over 10 seeds.

| Metr. | Transition | Expression |
|---|---|---|
| $SR_1$ | $0.89 \pm 0.05$ | $0.74 \pm 0.08$ |
| $SR_5$ | $0.99 \pm 0.01$ | $0.94 \pm 0.06$ |

**Language grounding phase** L→G: The LGG demonstrates the 5 types of generalization from Table 1. From known configurations, agents can generate more goals than they observed in training data (1, 2). They can do so from new initial configurations (3). They can generalize to new sentences (4) and even to combinations of new sentences and initial configurations (5). These results assert that DECSTR generalizes well in a variety of contexts and shows good behavioral diversity.

**Instruction following phase** L→G→B: Table 2 presents the 1-shot and 5-shot results in the *transition* and *expression* setups. In the sequence setups, DECSTR succeeds in $L = 14.9 \pm 5.7$ successive instructions (mean±stdev over 10 seeds). These results confirm efficient language grounding. DECSTR can follow instructions or sequences of instructions and generalize to their logical combinations. Strategy switching improves performance ($SR_5$ - $SR_1$). DECSTR also demonstrates strong behavioral diversity: when asked over 10 seeds to repeat 50 times the same instruction, it achieves at least 7.8 different configurations, 15.6 on average and up to 23 depending on the instruction.

## 4.2 DO WE NEED AN INTERMEDIATE REPRESENTATION?

This section investigates the need for an intermediate semantic representation. To this end, we introduce an end-to-end LC-RL baseline directly mapping Language to Behavior (L→B) and compare its performance with DECSTR in the instruction following phase (L→G→B).

**The LB baseline.** To limit the introduction of confounding factors and under-tuning concerns, we base this implementation on the DECSTR code and incorporate defining features of IMAGINE, a state-

of-the-art language conditioned RL agent (Colas et al., 2020). We keep the same HER mechanism, object-centered architectures and RL algorithm as DECSTR. We just replace the semantic goal space by the 102 language instructions. This baseline can be seen as an oracle version of the IMAGINE algorithm where the reward function is assumed perfect, but without the imagination mechanism.

**Comparison in the instruction following phase** L→B **vs** L→G→B: After training the LB baseline for 14K episodes, we compare its performance to DECSTR's in the instruction-following setup. In the *transition* evaluation setup, LB achieves $SR_1 = 0.76 \pm 0.001$: it always manages to move blocks close to or far from each other, but consistently fails to stack them. Adding more attempts does not help: $SR_5 = 0.76 \pm 0.001$. The LB baseline cannot be evaluated in the *expression* setup because it does not manipulate goal sets. Because it cannot stack blocks, LB only succeeds in $3.01 \pm 0.43$ random instructions in a row, against 14.9 for DECSTR (*sequence* setup). We then evaluate LB's diversity on the set of instructions it succeeds in. When asked to repeat 50 times the same instruction, it achieves at least 3.0 different configurations, 4.2 on average and up to 5.2 depending on the instruction against 7.8, 17.1, 23 on the same set of instructions for DECSTR. We did not observe *strategy-switching* behaviors in LB, because it either always succeeds (close/far instructions) or fails (stacks).

**Conclusion.** The introduction of an intermediate semantic representation helps DECSTR decouple skill learning from language grounding which, in turns, facilitates instruction-following when compared to the end-to-end language-conditioned learning of LB. This leads to improved scores in the *transition* and *sequence* setups. The direct language-conditioning of LB prevents the generalization to logical combination and leads to a reduced *diversity* in the set of mastered instructions. Decoupling thus brings significant benefits to LGB architectures.

## 4.3 DO WE NEED A SEMANTIC INTERMEDIATE REPRESENTATION?

This section investigates the need for the intermediate representation to be semantic. To this end, we introduce the LGB-C baseline that leverages continuous goal representations in place of semantic ones. We compare them on the two first phases.

**The LGB-C baseline.** The LGB-C baseline uses *continuous* goals expressing target block coordinates in place of semantic goals. The skill learning phase is thus equivalent to traditional goal-conditioned RL setups in block manipulation tasks (Andrychowicz et al., 2017; Colas et al., 2019a; Li et al., 2019; Lanier et al., 2019). Starting from the DECSTR algorithm, LGB-C adds a translation module that samples a set of target block coordinates matching the targeted semantic configuration which is then used as the goal input to the policy. In addition, we integrate defining features of the state-of-the-art approach from Lanier et al. (2019): non-binary rewards (+1 for each well placed block) and multi-criteria HER, see details in Appendix D.2.

**Comparison in skill learning phase** G→B: The LGB-C baseline successfully learns to discover and master all 35 semantic configurations by placing the three blocks to randomly-sampled target coordinates corresponding to these configurations. It does so faster than DECSTR: $708 \cdot 10^3$ episodes to reach $SR = 95\%$, against $1238 \cdot 10^3$ for DECSTR, see Appendix Figure 6. This can be explained by the denser learning signals it gets from using HER on continuous targets instead of discrete ones. In this phase, however, the agent only learns one parameterized skill: to place blocks at their target position. It cannot build a repertoire of semantic skills because it cannot discriminate between different block configurations. Looking at the sum of the distances travelled by the blocks or the completion time, we find that DECSTR performs opportunistic goal reaching: it finds simpler configurations of the blocks which satisfy its semantic goals compared to LGB-C. Blocks move less ($\Delta_{dist} = 26 \pm 5$ cm), and goals are reached faster ($\Delta_{steps} = 13 \pm 4$, mean$\pm$std across goals with p-values $> 1.3 \cdot 10^{-5}$ and $3.2 \cdot 10^{-19}$ respectively).

Table 3: LGB-C performance in the L→G phase. Mean over 10 seeds. Stdev $< 0.003$ and $0.008$ respectively.

| Metrics | Test 1 | Test 2 | Test 3 | Test 4 | Test 5 |
|---|---|---|---|---|---|
| Precision | 0.66 | 0.78 | 0.39 | 0.0 | 0.0 |
| Recall | 0.05 | 0.02 | 0.06 | 0.0 | 0.0 |

**Comparison in language grounding phase** L→G**:**   We train the LGG to generate continuous target coordinates conditioned on language inputs with a mean-squared loss and evaluate it in the same setup as DECSTR's LGG, see Table 3. Although it maintains reasonable precision in the first two testing sets, the LGG achieves low recall – i.e. diversity – on all sets. The lack of semantic representations of skills might explain the difficulty of training a language-conditioned goal generator.

**Conclusion.**   The skill learning phase of the LGB-C baseline is competitive with the one of DECSTR. However, the poor performance in the language grounding phase prevents this baseline to perform instruction following. For this reason, and because semantic representations enable agents to perform opportunistic goal reaching and to acquire repertoires for semantic skills, we believe the semantic representation is an essential part of the LGB architecture.

## 5   DISCUSSION AND CONCLUSION

This paper contributes LGB, a new conceptual RL architecture which introduces an intermediate semantic representation to decouple sensorimotor learning from language grounding. To demonstrate its benefits, we present DECSTR, a learning agent that discovers and masters all reachable configurations in a manipulation domain from a set of relational spatial primitives, before undertaking an efficient language grounding phase. This was made possible by the use of object-centered inductive biases, a new form of automatic curriculum learning and a novel language-conditioned goal generation module. Note that our main contribution is in the conceptual approach, DECSTR being only an instance to showcase its benefits. We believe that this approach could benefit from any improvement in GC-RL (for skill learning) or generative models (for language grounding).

**Semantic representations.**   Results have shown that using predicate-based representations was sufficient for DECSTR to efficiently learn abstract goals in an opportunistic manner. The proposed semantic configurations showcase promising properties: 1) they reduce the complexity of block manipulation where most effective works rely on a heavy hand-crafted curriculum (Li et al., 2019; Lanier et al., 2019) and a specific curiosity mechanism (Li et al., 2019); 2) they facilitate the grounding of language into skills and 3) they enable decoupling skill learning from language grounding, as observed in infants (Piaget, 1977). The set of semantic predicates is, of course, domain-dependent as it characterizes the space of behaviors that the agent can explore. However, we believe it is easier and requires less domain knowledge to define the set of predicates, i.e. the dimensions of the space of potential goals, than it is to craft a list of goals and their associated reward functions.

**A new approach to language grounding.**   The approach proposed here is the first simultaneously enabling to decouple skill learning from language grounding and fostering a diversity of possible behaviors for given instructions. Indeed, while an instruction following agent trained on goals like *put red close_to green* would just push the red block towards the green one, our agent can generate many matching goal configurations. It could build a pyramid, make a blue-green-red pile or target a dozen other compatible configurations. This enables it to *switch strategy*, to find alternative approaches to satisfy a same instruction when first attempts failed. Our goal generation module can also generalize to new sentences or transpose instructed transformations to unknown initial configurations. Finally, with the goal generation module, the agent can deal with any logical expression made of instructions by combining generated goal sets. It would be of interest to simultaneously perform language grounding and skill learning, which would result in "overlapping waves" of sensorimotor and linguistic development (Siegler, 1998).

**Semantic configurations of variable size.**   Considering a constant number of blocks and, thus, fixed-size configuration spaces is a current limit of DECSTR. Future implementations of LGB may handle inputs of variable sizes by leveraging Graph Neural Networks as in Li et al. (2019). Corresponding semantic configurations could be represented as a set of vectors, each encoding information about a predicate and the objects it applies to. These representations could be handled by Deep Sets (Zaheer et al., 2017). This would allow to target partial sets of predicates that would not need to characterize all relations between all objects, facilitating scalability.

**Conclusion**   In this work, we have shown that introducing abstract goals based on relational predicates that are well understood by humans can serve as a pivotal representation between skill learning

and interaction with a user through language. Here, the role of the social partner was limited to: 1) helping the agent to experience non-trivial configurations and 2) describing the agent's behavior in a simplified language. In the future, we intend to study more intertwined skill learning and language grounding phases, making it possible to the social partner to teach the agent during skill acquisition.

ACKNOWLEDGMENTS

This work was performed using HPC resources from GENCI-IDRIS (Grant 20XX-AP010611667), the MeSU platform at Sorbonne-Université and the PlaFRIM experimental testbed. Cédric Colas is partly funded by the French Ministère des Armées - Direction Générale de l'Armement.

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

## A    LGB PSEUDO-CODE

Algorithm 1 and 2 present the high-level pseudo-code of any algorithm following the LGB architecture for each of the three phases.

---

**Algorithm 1**  LGB architecture
G→B phase

    ▷ Goal → Behavior phase
1: **Require** Env $E$
2: Initialize policy $\Pi$, goal sampler $G_s$, buffer $B$
3: **loop**
4:    $g \leftarrow G_s$.sample()
5:    $(s, a, s', g, c_p, c'_p)_{\text{traj}} \leftarrow E$.rollout($g$)
6:    $G_s$.update($c_p^T$)
7:    $B$.update($(s, a, s', g, c_p, c'_p)_{\text{traj}}$)
8:    $\Pi$.update($B$)
9: **return** $\Pi, G_s$
10:
11:
12:

**Algorithm 2**  LGB architecture
L→G and L→G→B phases

    ▷ Language → Goal phase
1: **Require** $\Pi, E, G_s$, social partner $SP$
2: Initialize language goal generator $LGG$
3: dataset $\leftarrow SP$.interact($E, \Pi, G_s$)
4: $LGG$.update(dataset)
5: **return** $LGG$
    ▷ Language → Behavior phase
6: **Require** $E, \Pi, LGG, SP$
7: **loop**
8:    instr. $\leftarrow SP$.listen()
9:    **loop**    ▷ Strategy switching loop
10:      $g \leftarrow LGG$.sample(instr., $c^0$)
11:      $c_p^T \leftarrow E$.rollout($g$)
12:      **if** $g == c_p^T$ **then break**

---

## B    SEMANTIC PREDICATES AND APPLICATION TO FETCH MANIPULATE

In this paper, we restrict the semantic representations to the use of the *close* and *above* binary predicates applied to $M = 3$ objects. The resulting semantic configurations are formed by:

$$c_p = [c(o_1, o_2), c(o_1, o_3), c(o_2, o_3), a(o_1, o_2), a(o_2, o_1), a(o_1, o_3), a(o_3, o_1), a(o_2, o_3), a(o_3, o_2)],$$

where $c()$ and $a()$ refer to the *close* and *above* predicates respectively and $(o_1, o_2, o_3)$ are the red, green and blue blocks respectively.

**Symmetry and asymmetry of *close* and *above* predicates.**    We consider objects $o_1$ and $o_2$.

- *close* is symmetric: "$o_1$ is **close** to $o_2$" $\Leftrightarrow$ "$o_2$ is **close** to $o_1$". The corresponding semantic mapping function is based on the Euclidean distance, which is symmetric.
- *above* is asymmetric: "$o_1$ is **above** $o_2$" $\Rightarrow$ **not** "$o_2$ is **above** $o_1$". The corresponding semantic mapping function evaluates the sign of the difference of the object $Z$-axis coordinates.

## C    THE DECSTR ALGORITHM

### C.1    INTRINSICALLY MOTIVATED GOAL-CONDITIONED RL

**Overview.**    Algorithm 3 presents the pseudo-code of the sensorimotor learning phase (G→B) of DECSTR. It alternates between two steps:

- **Data acquisition.** A DECSTR agent has no prior on the set of reachable semantic configurations. Its first goal is sampled uniformly from the semantic configuration space. Using this goal, it starts interacting with its environment, generating trajectories of sensory states $s$, actions $a$ and configurations $c_p$. The last configuration $c_p^T$ achieved in the episode after $T$ time steps is considered stable and is added to the set of reachable configurations. As it interacts with the environment, the agent explores the configuration space, discovers reachable configurations and selects new targets.
- **Internal models updates.** A DECSTR agent updates two models: its curriculum strategy and its policy. The curriculum strategy can be seen as an active goal sampler. It biases the selection of goals to target and goals to learn about. The policy is the module controlling the agent's behavior and is updated via RL.

---

**Algorithm 3** DECSTR: sensorimotor phase G→B.

---

1: **Require:** env $E$, # buckets $N_b$, # episodes before biased init. $n_{\text{unb}}$, self-evaluation probability $p_{\text{self\_eval}}$, noise function $\sigma()$
2: **Initialize:** policy $\Pi$, buffer $B$, goal sampler $G_s$, bucket sampling probabilities $p_b$, language module $LGG$.
3: **loop**
4:      self\_eval ← random() $< p_{\text{self\_eval}}$                    ▷ If $True$ then evaluate competence
5:      $g \leftarrow G_s$.sample(self\_eval, $p_b$)
6:      biased\_init ← $epoch < n_{\text{unb}}$              ▷ Bias initialization only after $n_{\text{unb}}$ epochs
7:      $s^0, c_p^0 \leftarrow E$.reset($biased\_init$)           ▷ $c_0$: Initial semantic configuration
8:      **for** $t = 1 : T$ **do**
9:          $a^t \leftarrow policy(s^t, c^t, g)$
10:          **if not** self\_eval **then**
11:              $a^t \leftarrow a^t + \sigma()$
12:          $s^{t+1}, c_p^{t+1} \leftarrow E$.step($a^t$)
13:      episode ← $(s, c, a, s', c')$
14:      $G_s$.update($c^T$)
15:      $B$.update(episode)
16:      $g \leftarrow G_s$.sample($p_b$)
17:      batch ← $B$.sample($g$)
18:      $\Pi$.update(batch)
19:      **if** self\_eval **then**
20:          $p_b \leftarrow G_s$.update\_LP()

---

**Policy updates with a goal-conditioned Soft Actor-Critic.**  Readers familiar with Markov Decision Process and the use of SAC and HER algorithms can skip this paragraph.

We want the DECSTR agent to explore a semantic configuration space and master reachable configurations in it. We frame this problem as a goal-conditioned MDP (Schaul et al., 2015): $\mathcal{M} = (\mathcal{S}, \mathcal{G}_p, \mathcal{A}, \mathcal{T}, \mathcal{R}, \gamma)$, where the state space $\mathcal{S}$ is the usual sensory space augmented with the configuration space $\mathcal{C}_p$, the goal space $\mathcal{G}_p$ is equal to the configuration space $\mathcal{G}_p = \mathcal{C}_p$, $\mathcal{A}$ is the action space, $\mathcal{T} : \mathcal{S} \times \mathcal{A} \times \mathcal{S} \rightarrow [0, 1]$ is the unknown transition probability, $\mathcal{R} : \mathcal{S} \times \mathcal{A} \rightarrow \{0, 1\}$ is a sparse reward function and $\gamma \in [0, 1]$ is the discount factor.

Policy updates are performed with Soft Actor-Critic (SAC) (Haarnoja et al., 2018), a state-of-the-art off-policy actor-critic algorithm. We also use Hindsight Experience Replay (HER) (Andrychowicz et al., 2017). This mechanism enables agents to learn from failures by reinterpreting past trajectories in the light of goals different from the ones originally targeted. HER was designed for continuous goal spaces, but can be directly transposed to discrete goals (Colas et al., 2019a). In our setting, we simply replace the originally targeted goal configuration by the currently achieved configuration in the transitions fed to SAC. We also use our automatic curriculum strategy: the LP-C-based probabilities are used to sample goals to learn about. When a goal $g$ is sampled, we search the experience buffer for the collection of episodes that ended in the configuration $c_p = g$. From these episodes, we sample a transition uniformly. The HER mechanism substitutes the original goal with one of the configurations achieved later in the trajectory. This substitute $g$ has high chances of being the sampled one. At least, it is a configuration on the path towards this goal, as it is sampled from a trajectory leading to it. The HER mechanism is thus biased towards goals sampled by the agent.

**Object-Centered Inductive Biases.**  In the proposed *Fetch Manipulate* environment, the three blocks share the same set of attributes (position, velocity, color identifier). Thus, it is natural to encode a *relational inductive bias* in our architecture. The behavior with respect to a pair of objects should be independent from the position of the objects in the inputs. The architecture used for the policy is depicted in Figure 4.

A shared network ($NN_{\text{shared}}$) encodes the concatenation of: 1) agent's body features; 2) object pair features; 3) current configuration ($c_p$) and 4) current goal $g$. This is done independently for all object pairs. No matter the location of the features of the object pair in the initial observations, this shared network ensures that the same behavior will be performed, thus skills are transferred between object

pairs. A sum is then used to aggregate these outputs, before a final network ($NN_{\text{policy}}$) maps the aggregation to actions $a$. The critic follows the same architecture, where a final network $NN_{\text{critic}}$ maps the aggregation to an action-value $Q$. Parallel encoding of each pair-specific inputs can be seen as different modules trying to reach the goal by only seeing these pair-specific inputs. The intuition is that modules dealing with the pair that should be acted upon to reach the goal will supersede others in the sum aggregation.

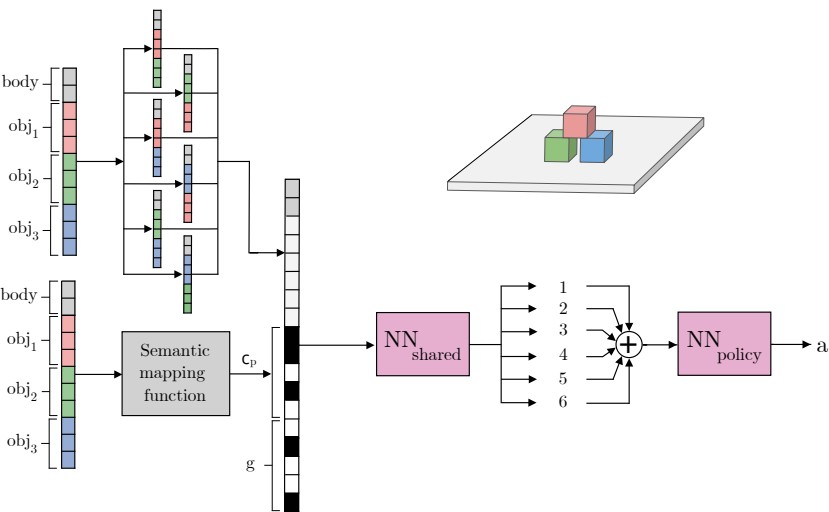

Figure 4: Object-centered modular architecture for the policy.

Although in principle our architecture could work with combinations of objects (3 modules), we found permutations to work better in practice (6 modules). With combinations, the shared network would need to learn to put block $A$ on block $B$ to achieve a predicate $above(o_i, o_j)$, and would need to learn the reverse behavior (put $B$ on $A$) to achieve the symmetric predicate $above(o_j, o_i)$. With permutations, the shared network can simply learn one of these behaviors (e.g. $A$ on $B$). Considering the predicate $above(o_A, o_B)$, at least one of the modules has objects organized so that this behavior is the good one: if the permutation $(o_B, o_A)$ is not the right one, permutation $(o_A, o_B)$ is. The symmetry bias is explained in Section 3.4. It leverages the symmetry of the behaviors required to achieve the predicates $above(o_i, o_j)$ and $above(o_j, o_i)$. As a result, the two goal configurations are:

$$g_1 = [c(o_1,o_2),\ c(o_1,o_3),\ c(o_2,o_3),\ a(o_1,o_2),\ a(o_1,o_3),\ a(o_2,o_3)],$$

$$g_2 = [c(o_1,o_2),\ c(o_1,o_3),\ c(o_2,o_3),\ a(o_2,o_1),\ a(o_3,o_1),\ a(o_3,o_2)],$$

where $g_1$ is used in association with object permutations $(o_i, o_j)$ with $i < j$ and $g_2$ is used in association with object permutations $(o_j, o_i)$ with $i < j$. As a result, the shared network automatically ensures transfer between predicates based on symmetric behaviors.

**Implementation Details.** This part includes details necessary to reproduce results. The code is available at https://sites.google.com/view/decstr/.

*Parallel implementation of* SAC-HER. We use a parallel implementation of SAC (Haarnoja et al., 2018). Each of the 24 parallel worker maintains its own replay buffer of size $10^6$ and performs its own updates. Updates are summed over the 24 actors and the updated network are broadcast to all workers. Each worker alternates between 2 episodes of data collection and 30 updates with batch size 256. To form an epoch, this cycle is repeated 50 times and followed by the offline evaluation of the agent on each reachable goal. An epoch is thus made of $50 \times 2 \times 24 = 2400$ episodes.

*Goal sampler updates.* The agent performs self-evaluations with probability $self\_eval = 0.1$. During these runs, the agent targets uniformly sampled discovered configurations without exploration noise. This enables the agent to self-evaluate on each goal. Goals are organized into buckets. Main Section 3.4 presents our automatic bucket generation mechanism. Once buckets are formed, we compute $C$, $LP$ and $P$, based on windows of the past $W = 1800$ self-evaluation interactions for each bucket.

*Modular architecture.* The shared network of our modular architecture $NN_{\text{shared}}$ is a 1-hidden layer network of hidden size 256. After all pair-specific inputs have been encoded through this module, their output (of size 84) are summed. The sum is then passed through a final network with a hidden layer of size 256 to compute the final actions (policy) or action-values (critic). All networks use $ReLU$ activations and the Xavier initialization. We use Adam optimizers, with learning rates $10^{-3}$. The list of hyperparameters is provided in Table 4.

Table 4: Sensorimotor learning hyperparameters used in DECSTR.

| Hyperparam. | Description | Values. |
|---|---|---|
| $nb\_mpis$ | Number of workers | 24 |
| $nb\_cycles$ | Number of repeated cycles per epoch | 50 |
| $nb\_rollouts\_per\_mpi$ | Number of rollouts per worker | 2 |
| $nb\_updates$ | Number of updates per cycle | 30 |
| $start\_bias\_init$ | Epoch from which initializations are biased | 100 |
| $W$ | Curriculum window size | 1800 |
| $self\_eval$ | Self evaluation probability | 0.1 |
| $N_b$ | Number of buckets | 5 |
| $replay\_strategy$ | HER replay strategy | $future$ |
| $k\_replay$ | Ratio of HER data to data from normal experience | 4 |
| $batch\_size$ | Size of the batch during updates | 256 |
| $\gamma$ | Discount factor to model uncertainty about future decisions | 0.98 |
| $\tau$ | Polyak coefficient for target critics smoothing | 0.95 |
| $lr\_actor$ | Actor learning rate | $10^{-3}$ |
| $lr\_critic$ | Critic learning rate | $10^{-3}$ |
| $\alpha$ | Entropy coefficient used in SAC | 0.2 |
| $automatic\_entropy$ | Automatically tune the entropy coefficient | $False$ |

**Computing resources.** The sensorimotor learning experiments contain 8 conditions: 2 of 10 seeds and 6 of 5 seeds. Each run leverages 24 cpus (24 actors) for about 72h for a total of 9.8 cpu years. Experiments presented in this paper requires machines with at least 24 cpu cores. The language grounding phase runs on a single cpu and trains in a few minutes.

## C.2  LANGUAGE-CONDITIONED GOAL GENERATOR

**Language-Conditioned Goal Generator Training.** We use a conditional Variational Auto-Encoder (C-VAE) (Sohn et al., 2015). Conditioned on the initial configuration and a sentence describing the expected transformation of one object relation, it generates compatible goal configurations. After the first phase of goal-directed sensorimotor training, the agent interacts with a hard-coded social partner as described in Main Section 3. From these interactions, we obtain a dataset of 5000 triplets: initial configuration, final configuration and sentence describing one change of predicate from the initial to the final configuration. The list of sentences used by the synthetic social partner is provided in Table 5. Note that *red*, *green* and *blue* refer to objects $o_1$, $o_2$, $o_3$ respectively.

**Content of test sets.** We describe the 5 test sets:

1. Test set 1 is made of input pairs $(c_i, s)$ from the training set, but tests the coverage of all compatible final configurations $\mathcal{C}_f$, 80% of which are not found in the training set. In that sense, it is partly a test set.
2. Test set 2 contains two input pairs: $\{[0\ 1\ 0\ 0\ 0\ 0\ 0\ 0\ 0]$, *put blue close_to green*$\}$ and $\{[0\ 0\ 1\ 0\ 0\ 0\ 0\ 0\ 0]$, *put green below red*$\}$ corresponding to 7 and 24 compatible final configurations respectively.
3. Test set 3 corresponds to all pairs including the initial configuration $c_i = [1\ 1\ 0\ 0\ 0\ 0\ 0\ 0\ 0]$ (29 pairs), with an average of 13 compatible final configurations.
4. Test set 4 corresponds to all pairs including one of the sentences *put green on_top_of red* and *put blue far_from red*, i.e. 20 pairs with an average of 9.5 compatible final configurations.

5. Test set 5 is all pairs that include both the initial configuration of test set 3 and one of the sentences of test set 4, i.e. 2 pairs with 6 and 13 compatible goals respectively. Note that pairs of set 5 are removed from sets 3 and 4.

Table 5: List of instructions. Each of them specifies a shift of one predicate, either from false to true $(0 \rightarrow 1)$ or true to false $(1 \rightarrow 0)$. **block A** and **block B** represent two different blocks from {red, blue, green}.

| Transition type | Sentences |
|---|---|
| Close $0 \rightarrow 1$ ($\times 3$) | *Put **block A** close_to **block B**, Bring **block A** and **block B** together,* *Get **block A** and **block B** close_from each_other, Get **block A** close_to **block B**.* |
| Close $1 \rightarrow 0$ ($\times 3$) | *Put **block A** far_from **block B**, Get **block A** far_from **block B**,* *Get **block A** and **block B** far_from each_other, Bring **block A** and **block B** apart,* |
| Above $0 \rightarrow 1$ ($\times 6$) | *Put **block A** above **block B**, Put **block A** on_top_of **block B**,* *Put **block B** under **block A**, Put **block B** below **block A**.* |
| Above $1 \rightarrow 0$ ($\times 6$) | *Remove **block A** from_above **block B**, Remove **block A** from **block B**,* *Remove **block B** from_below **block A**, Put **block B** and **block A** on_the_same_plane,* *Put **block A** and **block B** on_the_same_plane.* |

**Testing on logical expressions of instructions.** To evaluate DECSTR on logical functions of instructions, we generate three types of expressions:

1. 100 instructions of the form "A and B" where A and B are basic instructions corresponding to shifts of the form *above* $0 \rightarrow 1$ (see Table 5). These intersections correspond to stacks of 3 or pyramids.
2. 200 instructions of the form "A and B" where A and B are *above* and *close* instructions respectively. B can be replaced by "not B" with probability 0.5.
3. 200 instructions of the form "(A and B) or (C and D))", where A, B, C, D are basic instructions: A and C are *above* instructions while B and D are *close* instructions. Here also, any instruction can be replaced by its negation with probability 0.5.

**Implementation details.** The encoder is a fully-connected neural network with two layers of size 128 and *ReLU* activations. It takes as input the concatenation of the final binary configuration and its two conditions: the initial binary configuration and an embedding of the NL sentence. The NL sentence is embedded with an recurrent network with embedding size 100, *tanh* non-linearities and biases. The encoder outputs the mean and log-variance of the latent distribution of size 27. The decoder is also a fully-connected network with two hidden layers of size 128 and *ReLU* activations. It takes as input the latent code $z$ and the same conditions as the encoder. As it generates binary vectors, the last layer uses *sigmoid* activations. We train the architecture with a mixture of Kullback-Leibler divergence loss $(KD_{\text{loss}})$ w.r.t a standard Gaussian prior and a binary Cross-Entropy loss $(BCE_{\text{loss}})$. The combined loss is $BCE_{\text{loss}} + \beta \times KD_{\text{loss}}$ with $\beta = 0.6$. We use an Adam optimizer, a learning rate of $5 \times 10^{-4}$, a batch size of 128 and optimize for 150 epochs. As training is fast ($\approx$ 2 min on a single cpu), we conducted a quick hyperparameter search over $\beta$, layer sizes, learning rates and latent sizes (see Table 6). We found robust results for various layer sizes, various $\beta$ below 1. and latent sizes above 9.

Table 6: LGG hyperparameter search. In bold are the selected hyperparameters.

| Hyperparam. | Values. |
|---|---|
| $\beta$ | [0.5, **0.6**, 0.7, 0.8, 0.9, 1.] |
| layers size | [**128**, 256] |
| learning rate | [0.01, **0.005**, 0.001] |
| latent sizes | [9, 18, **27**] |

## D  BASELINES AND ORACLE

The language-conditioned LB baseline is fully described in the main document.

### D.1 EXPERT BUCKETS ORACLE

In the EXPERT BUCKETS oracle, the automatic bucket generation of DECSTR is replaced with an expert-predefined set of buckets using *a priori* measures of similarity and difficulty. To define these buckets, one needs prior knowledge of the set of unreachable configurations, which are ruled out. The 5 predefined buckets contain all configurations characterized by:

- Bucket 1: a single *close* relation between a pair of objects and no *above* relations (4 configurations).
- Bucket 2: 2 or 3 *close* relations and no *above* relations (4 configurations).
- Bucket 3: 1 stack of 2 blocks and a third block that is either away or close to the base, but is not close to the top of the stack (12 configurations).
- Bucket 4: 1 stack of 2 blocks and the third block close to the stack, as well as pyramid configurations (9 configurations).
- Bucket 5: stacks of 3 blocks (6 configurations).

These buckets are the only difference between the EXPERT BUCKETS baseline and DECSTR.

### D.2 LGB-C BASELINE

The LGB-C baseline represent goals not as semantic configurations but as particular 3D targets positions for each block, as defined for example in Lanier et al. (2019) and Li et al. (2019). The goal vector size is also 9 and contains the 3D target coordinates of the three blocks. This baselines also implements decoupling and, thus, can be compared to DECSTR in the three phases. We keep as many modules as possible common with DECSTR to minimize the amount of confounding factors and reduce the *under-fitting* bias. The goal selection is taken from DECSTR, but converts semantic configuration into specific randomly-sampled target coordinates for the blocks, see Figure 5. The agent is not conditioned on its current semantic configuration nor its semantic goal configuration. For this reason, we do not apply the symmetry bias. The binary reward is positive when the maximal distance between a block and its target position is below 5 cm, i.e. the size of a block (similar to (Andrychowicz et al., 2017)). To make this baseline competitive, we integrate methods from a state of the art block manipulation algorithm (Lanier et al., 2019). The agent receives positive rewards of 1, 2, 3 when the corresponding number of blocks are well placed. We also introduce the multi-criteria HER from Lanier et al. (2019). Finally, we add an additional object-centered inductive bias by only considering, for each Deep Sets module, the 3D target positions of the corresponding pair. That is, for each object pair, we ignore the 3D positions of the remaining object, yielding to a vector of size 6. Language grounding is based on a C-VAE similar to the one used by DECSTR. We only replace the cross-entropy loss by a mean-squared loss due to the continuous nature of the target goal coordinates. We use the exact same training and testing sets as with semantic goals.

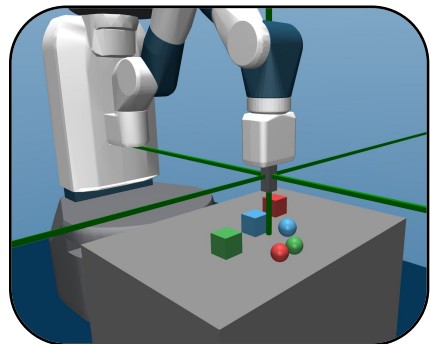

Figure 5: The LGB-C baseline samples target positions for each block (example for a pyramid here).

# E  ADDITIONAL RESULTS

## E.1  COMPARISON DECSTR - LGB-C IN SKILL LEARNING PHASE

Figure 6 presents the average success rate over the 35 valid configurations during the skill learning phase for DECSTR and the LGB-C baseline. Because LGB-C cannot pursue semantic goals as such, we randomly sample a specific instance of this semantic goal: target block coordinates that satisfy the constraints expressed by it. Because LGB-C is not aware of the original semantic goal, we cannot measure success as the ability to achieve it. Instead, *success* is defined as the achievement of the corresponding specific goal: bringing blocks to their respective targets within an error margin of 5 cm each. In short, DECSTR targets semantic goals and is evaluated on its ability to reach them. LGB-C targets specific goals and is evaluated on its ability to reach them. These two measures do not match exactly. Indeed, LGB-C sometimes achieves its specific goal but, because of the error margins, does not achieve the original semantic goal.

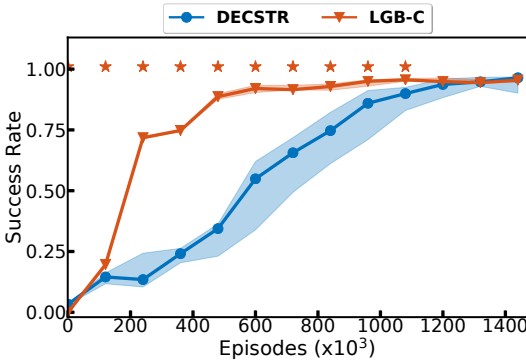

Figure 6: Comparison DECSTR and LGB-C in the skill learning phase.

## E.2  AUTOMATIC BUCKET GENERATION.

Figure 7 depicts the evolution of the content of buckets along training (epochs 1, 50 and 100). Each pie chart corresponds to a reachable configuration and represents the distribution of configurations into buckets across 10 different seeds. Blue, orange, green, yellow, purple represent buckets 1 to 5 respectively and grey are undiscovered configurations. At each moment, the discovered configurations are equally spread over the 5 buckets. A given configuration may thus change bucket as new configurations are discovered, so that the ones discovered earlier are assigned buckets with lower indexes. Goals are organized by their bucket assignments in the *Expert Buckets* condition (from top to bottom).

After the first epoch (left), DECSTR has discovered all configurations from the expert buckets 1 and 2, and some runs have discovered a few other configurations. After 50 epochs, more configurations have been discovered but they are not always the same across runs. Finally, after 100 epochs, all configurations are found. Buckets are then steady and can be compared to expert-defined buckets. It seems that easier goals (top-most group) are discovered first and assigned in the first-easy buckets (blue and orange). Hardest configurations (stacks of 3, bottom-most group) seem to be discovered last and assigned the last-hardest bucket (purple). In between, different runs show different compositions, which are not always aligned with expert-defined buckets. Goals from expert-defined buckets 3 and 4 (third and fourth group from the top) seem to be attributed different automatic buckets in different runs. This means that they are discovered in different orders depending on the runs. In summary, easier and harder goals from expert buckets 1 - 2 and 5 respectively seem to be well detected by our automatic bucket generations. Goals in medium-level expected difficulty as defined by expert buckets seem not to show any significant difference in difficulty for our agents.

## E.3  DECSTR LEARNING TRAJECTORIES

Figure 8 shows the evolution of internal estimations of the competence C, the learning progress LP and the associated sampling probabilities P. Note that these metrics are computed online by

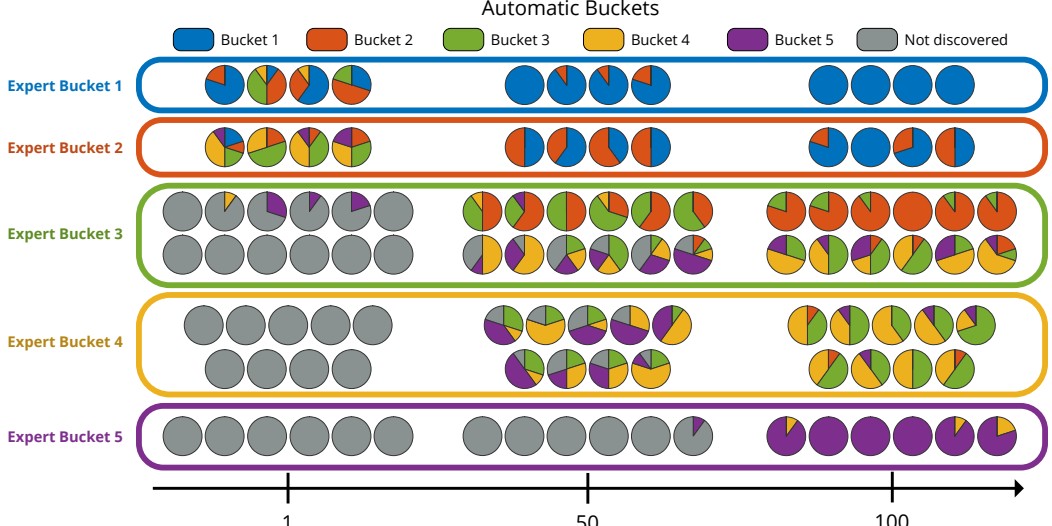

Figure 7: Evolution of the content of buckets from automatic bucket generation: epoch 1 (2400 episodes, left), 50 (middle) and 100 (right). Each pie chart corresponds to one of the 35 valid configurations. It represents the distribution of the bucket attributions of that configuration across 10 runs. Blue, orange, green, yellow, purple represent automatically generated buckets 1 to 5 respectively (increasing order of difficulty) and grey represents undiscovered configurations. Goals are organized according to their expert bucket attributions in the *Expert Buckets* condition (top-bottom organization).

DECSTR, as it self-evaluates on random discovered configurations. Learning trajectories seem to be uniform across different runs, and buckets are learned in increasing order. This confirms that the time of discovery is a good proxy for goal difficulty. In that case, configurations discovered first end up in the lower index buckets and are indeed learned first. Note that a failing automatic bucket generation would assign goals to random buckets. This would result in uniform measures of learning progress across different buckets, which would be equivalent to uniform goal sampling. As Main Figure 3c shows, DECSTR performs much better than the *random goals* conditions. This proves that our automatic bucket algorithm generates useful goal clustering.

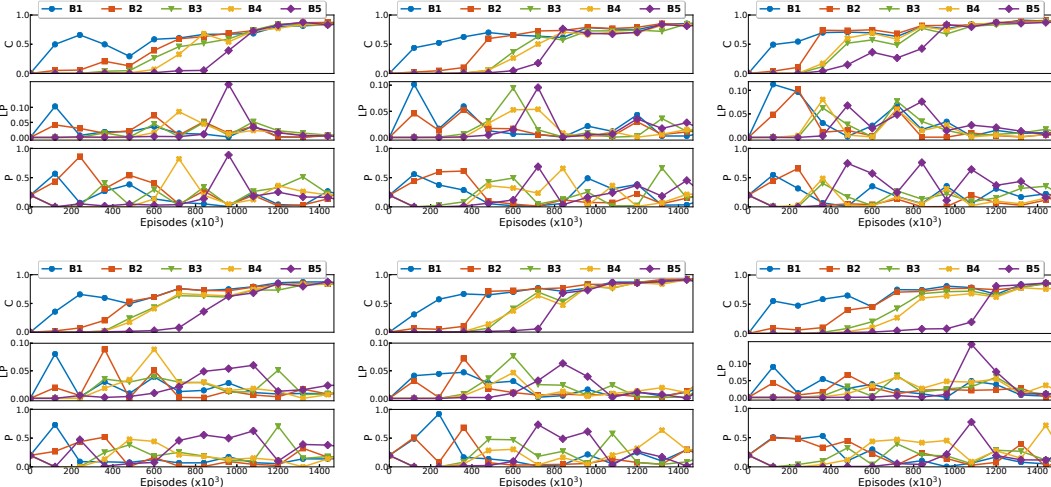

Figure 8: Learning trajectories of 6 DECSTR agents.

