# OpenReview forum: "Grounding Language to Autonomously-Acquired Skills via Goal Generation"
_ICLR.cc/2021/Conference — ICLR 2021 Poster_

### Official Review · AnonReviewer3 · 2020-10-19
**The paper presents a method for two-stage self-supervised RL, where an agent first acquires semantic concepts and second grounds language tokens to these concepts (object relationships, primarily). While motivated by language grounding, the training and evaluation paradigms do not include any natural language input from human users, and object relationships are limited to two binary predicates between three objects.**

**Rating:** 7
**Confidence:** 3

**Review:**


The DECSTR system's intrinsic motivations may be applicable to other application domains, depending on how objects and relations are enumerated. This potential is not explored beyond the toy environment presented. The learning methods (especially inductive biases) are hand-crafted based on human-level knowledge about semantic predicates, but only two ("above" and "close") are demonstrated. Without demonstrating the system on any other configuration or world, it's difficult to tell whether it's able to solve only the problem it's been crafted to solve in this specific environment.

Questions:

3.1 "in principle ... could use any other combination of binary predicates and could be extended to use n-ary predicates" this claim is not demonstrated in the paper, and in 3.2 the inductive biases seem bespoke crafted for binary predicate 'above' which has particular symmetry. Would similar careful design of inductive biases be necessary and possible for n-ary predicates that do not demonstrate these as easily (e.g., "topmost")? What about predicates that involve an unspecified number of discrete arguments, like "base" -> holding up an indefinite N of other objects/structures in "use the green block as the base".

3.4 "or is union" this doesn't generally hold for natural language. A statement like "put the red block or the green block above the yellow block" does not mean to put both red and green (union of goals) above yellow. Typically langauge "or" is "xor"; is the notion of "or" here not given in language or not meant to represent human language?


Areas for Improvement:

5 "a learning architecture that discovers and masters all reachable configurations from a set of relational primitives" this is literally true but only demonstrated on 'a' single set of relational primitives, so it feels like overclaiming.


Nits:
- double citation for Mandler, 2012 in intro in adjacent sentences can be condensed to once
- footnotes on other side of period
- "Besides" in "Blocks Manipulation" seems a bit off-sounding; maybe "In addition,"?
- typo section 3 "based o abstract"
- Typo section 5 backwards quotes "overlapping waves" LHS.
- "Caregiver" in section 5 is an unintroduced role. The rest of the paper does not frame DECSTR or the oracle generator this way.
- Ending the paper with "etc." feels weird/informal.

---

> ### Author Response · Authors · 2020-11-19
> **Answer to Reviewer 3**
>
> We thank Reviewer 3 for their helpful feedback.  Here, we answer comments that were specific to Reviewer 3. Concerns shared with other reviewers are addressed in the general answer.
>
> **Application to other domains**
>
> We partly answer this point in the main answer. The definition of semantic representations is domain-specific just like the definition of goal spaces and reward functions in traditional goal-conditioned RL. Instead of defining the set of tasks, we define the set of possible behaviors by defining the dimensions of that space. In the main answer, we argue that it is an easier task that involves less prior knowledge.
>
> From a developmental point of view, such sensors could be innate or acquired very early by infants, as it is the case for spatial predicates (Mandler, 2012). In future work, we would like to investigate how to learn these predicates from social interactions. The question of learning semantic predicates that can be used across a large variety of domains is also very interesting.
>
> However, the main contribution of this paper is to define and demonstrate the benefits of the decoupled LGB architecture over standard language-conditioned RL approaches. In this demonstration, DECSTR provides an illustration of the three properties emerging from LGB architectures (see main answer). Thus, the design/learning of semantic representation that generalize across domains and allow to represent a diversity of interesting behaviors is orthogonal to the main contribution of the paper.
>
> **About the use of more complicated predicates**
>
> "in principle ... could use any other combination of binary predicates and could be extended to use n-ary predicates": by this sentence, we mean that semantic representations can be composed of any combinations of n-ary predicates that we can think of. Of course, more complicated semantic representations also involve more complicated learning architectures to handle them. We clarified this sentence in the new version.
>
> About the “above” inductive bias: we can argue that infants who have an innate sensor for the “above” relation might also have the innate implicit knowledge of the symmetry of that relation.
>
> About other predicates: the sentence “use the green block as the base” could be seen as describing many binary “above” relations where the green block is always the block below. “Top-most” could also use a binary predicate “generally above” that does not require horizontal alignment, it would then refer to the block that is never the block below in all the existing “generally above” relations. Overall, inductive biases are not theoretically required to handle n-ary predicates, but they are practically useful for our current algorithms to work with reasonable sample sizes. Future implementations of the LGB architecture might require the use of Graph Neural Networks to handle relations between several nodes-objects.
>
> **About logical combinations of instructions**
>
> Reviewer 3 is right, our logical combinations are symbolic and are not expressed directly by text. It can be seen as assuming that the agent has an innate knowledge of the OR, AND and NOT logical functions, and can use them to combine any atomic instructions it discovered during its interaction with the tutor. How to translate complicated sentences that express logical combinations into logical trees of basic instructions that the agent could handle is very interesting but out of the scope of this present work.
>
> **"a learning architecture that discovers and masters all reachable configurations from a set of relational primitives"**
>
> We acknowledge that this formulation could be misinterpreted and corrected it.
>
> **Typos**
>
> We thank Reviewer 3 for pointing minor errors in the text. We corrected them in the new version.
>
> **Conclusion**
>
> Reviewer 3 seems mostly concerned about the generality of our approach and its use in other domains. With our new positioning detailed in the main answer, we argue that this paper presents a general architecture (LGB) and that DECSTR is only a particular implementation of it, to demonstrate its benefits. The design of semantic representations is orthogonal to the approach of this paper but is an interesting topic for future research. Designing semantic representation is defining the space of behaviors that the agent can explore. Although it is simpler than designing space of achievable goals and their associated rewards, it remains domain dependent. Designing or learning general predicates, and designing learning architecture able to handle complicated n-ary predicates is out of the scope of this paper.

---

> > ### Comment · AnonReviewer3 · 2020-11-24
> > **Reply**
> >
> > Thanks for these clarifications and those in the general response. I like the new framing of the paper and I think the contribution is much more clearly scoped.

---

### Official Review · AnonReviewer2 · 2020-10-26
**An interesting approach, limited by domain-specificity and poor paper organization**

**Rating:** 6
**Confidence:** 3

**Review:**

This work proposed DECSTR, a procedure for encouraging intrinsic motivation via an intermediate semantic state-space representation. The authors propose an intermediate semantic state space that the intrinsically motivated agent learns to explore. For the environment provided (a 3-block system), the agent fully explores the symbolic state space, reaching all feasible symbolic states. In the second part of the work, the authors train a language model capable of proposing symbolic goals (in the form of symbolic states) from natural language input and shows that the previously-intrinsically-motivated agent can now be made to reach these goals, demonstrating that the symbolic-goal-conditioned policy is sufficient for instruction following in their 3-block domain.

The work is generally interesting, and seems to address a simple version of a broader class of problems that embodied agents typically struggle with, particularly in the absence of clear goals. However, the approach presented in the behavior (in particular the form of the semantic representation that is claimed as one of the primary contributions of the work) is very specific to the single problem used for demonstrations in the paper, limiting the potential impact of the work.

First---and I think the most significant issue with the submission---is that many critical experimental details are included only in the lengthy appendix. Much of this information, including the information provided to the learning algorithm at every step and how that information is encoded such that it allows for a relatively object-agnostic representation, is only available in sufficient detail in the appendix. Relatedly, visualizations of the approach and experimental setup also only appear in the appendix, yet are extremely helpful (if not essential) for understanding. Detail critical to understanding the approach should be included in the body of the text.

Second, it is unclear exactly what problem is being solved in this work or what its primary contribution is. A clearer statement of its motivations will be necessary before publication. /What problem is the robot or system designers trying to overcome?/ Right now, the paper seems to come up with three potential answers to this question, none of which necessarily rises above the others. Here are what I think the main contributions of the work could be:

1. *The proposed semantic representation* The semantic goal representation used to define the space of intrinsic motivation seems to be a novel contribution. However, if the paper were to focus on this aspect of the contribution, it would need to do a better job understating why this representation were useful beyond a relatively small manipulation task. Critically: using only one problem setting with only three blocks is insufficient to convince the reader that this representation is useful more generally (as might be suggested by much of the talk about Inductive Bias).
2. *State of the art state-space exploration in intrinsic motivation.* This might be true, though I find such a thing hard to measure. In addition, it seems that many if not all of the tools used in the learning process are not novel. (Perhaps a combination of this and point 1. is the primary contribution.)
3. *State of the art performance on language-driven block manipulation tasks.* This might be true as well, but the results are so-far unconvincing. All baselines are varied forms of the proposed agent, which makes it difficult to compare against other approaches (e.g. something like Li et al 2019).

The paper currently seems to claim that the combination of progress in these three areas is a novel contribution; I am sympathetic to this idea (as I do not believe that every paper needs to be "state of the art" in one single thing), though it is sufficiently unclear at the moment what the takeaway message of the paper is that I cannot recommend it be published in its current state. In particular, the authors need to work on honing the message of the paper. It is also not unlikely that one or two more experiments will need to be added to support the focused narrative.

Smaller comments
- The name of algorithm should appear in the body of the text, not a footnote. Relatedly, it is unclear how the proposed approach uses the "Deep Sets" work in such a way that it justifies inclusion in the name of the proposed technique.
- The paper/Introduction would benefit from a summary of contributions: even after reading, it may not be clear to a reader which contributions are from this paper versus other work.
- Relatedly, much of the discussion of Inductive Biases that appear throughout the paper is of mixed relevance for this work. On the one hand, it is clear how the idea of an object-centric inductive bias helped to inform how the input to the neural network was encoded in a way that might allow the agent to apply its knowledge learned between two of the objects to a policy that allows it to manipulate all three. However, the goal condition is necessarily specific when it comes to representing which objects to which each element it refers. The structure of the goal and the semantic relations it encodes are quite specific to the particular problem at hand, and it is
- The reward for the "Position only" baseline seems artificially constructed: a non-binary reward function would likely allow the system to learn more easily. As of now I am unconvinced that the authors have worked hard enough to make a fair baseline for comparison. This is particularly problematic since this baseline is a key motivator for the existence of the proposed semantic goal representation.
- The paper overall is quite well written, despite relegating too much information to the abstracts.

---

> ### Author Response · Authors · 2020-11-19
> **Answer to Reviewer 2**
>
> We thank Reviewer 2 for their helpful feedback.  Here, we answer comments that were specific to Reviewer 2. Concerns shared with other reviewers are addressed in the general answer.
>
> **Comments on contributions and experimental section**
>
> Reviewer 2 seems to be mostly concerned about the lack of focus of our paper. We thank R2 for their very constructive suggestions on that aspect. These helped us refocus our paper as detailed in the main answer. We believe the organization of the experimental section is drastically improved by the new positioning of the paper. We also moved important information and visualisations back from the appendix to the main document.
>
> **Generality of the semantic representation**
>
> We answer this comment in the main answer.
>
> **Smaller comments**
>
> * DECSTR is now the name of the particular instance of the general LGB architecture we propose. This particular implementation relies on Deep Sets, which can justify the use of the term in the name. However, we removed “DECSTR” from the title of the paper, as it is only a secondary contribution, the LGB architecture being the central contribution.
> * We now explicitly list our contributions in the introduction
> * The description of inductive biases is indeed secondary given the new focus of the paper on the general LGB architecture. The methods section is updated accordingly.
> * We thank Reviewer 2 for a comment that pushed to allocate extra resources to the position baseline. We reached higher performance by combining non-binary rewards (as advised by Reviewer 2) and the multi-criteria HER method from Lanier et al., 2019. This helps get the Position baseline closer to state-of-the-art RL approaches for manipulation tasks.
>
> **Conclusion**
>
> It seems the main concerns of Reviewer 2 are about the contribution and challenge statements as well as the organization of the experimental section. We hope that our answers and the new version of the paper help resolve them.

---

### Official Review · AnonReviewer4 · 2020-10-29
**Well motivated, sum greater than its parts, but some concerns with baselines**

**Rating:** 6
**Confidence:** 5

**Review:**

**Summary**
This paper proposes DECSTR, a goal-driven RL framework where the goal is represented as a binary vector that encodes the semantic relationships between objects. The state is assumed to contain disentangled features for each of the objects (and other features relating to the agent’s end-effectors). The architecture is based on Deep Sets (Zaher et al., 2017), which allows the pairs of the objects to be encoded with a shared network. The paper also introduces a curriculum learning strategy similar to CURIOUS (Colas et al., 2019), which relies on metrics such as competence and learning progress (LP) in order to select goals to pursue during an episode. One key difference is that unlike CURIOUS which uses expert-defined “goal buckets”, DECSTR groups the goals based on recency of discovery. Once trained to be able to behave with respect to these semantic relationship goals, the second phase is language grounding. They learn a module (implemented as C-VAE) that converts from natural language text to the semantic configuration goal space. Experiments were conducted in the Fetch Manipulate robotic arm environment and compared with ablations of DECSTR without some of its components, demonstrating strong performance and generalization to various types of language instructions.

**Pros**:
- The paper is well-motivated, citing literature from several fields.
- The sum is greater than its parts: many components in DECSTR are based on existing works (e.g. Deep Sets, C-VAE, using LP for intrinsically motivated goals, etc.), but empirically they have shown through ablations that all of their components were necessary for the agent to solve the Fetch Manipulation task successfully.
- The experiment sections are fairly thorough, with ablations on the components of their methods (as said above), and various kinds of language command generalization evaluations (in a similar style to IMAGINE (Colas et al., 2020).
- The interpretability of the semantic goal space aspect is interesting. And being able to have the agent explicitly maps from the natural language text to the semantic goal space also helps us debug/understand what the agent is thinking at inference time

**Cons**:
- Part of the thesis is that decoupling of sensorimotor learning from language acquisition is advantageous to an end-to-end language to sensorimotor learning. I have concerns/clarification about some of the baselines, which might not have been a fair comparison with DECSTR (see question 1 & 2 below)
- Some parts of the method are unclear/vague without reading the appendix section to get the full detail. I understand that is due to the space limitation issue and because there are so many components to DECSTR. (see question 3)

**Recommendation**:

Overall, I vote for marginally below acceptance threshold in the current form. As mentioned in the strengths section, I do like the motivation of the paper and the strong performance of the method. But I am also suspicious of the poor performance of the baselines (e.g. Figure 1c), which may be due to not having HER, instead of their proposed contributions. It would be good if the authors can clarify that concern.

**Question**:
1. In Figure 1c, for the Language Goals baseline, was HER applied to the Language Goals in this case (i.e. similar to ACTRCE (Chan et al., 2019), IMAGINE (Colas et al., 2020)? Similarly, was HER applied to the Position-Goals baseline? If not, then it is possible the difference in performance between DECSTR and these baselines may be due more to HER than due to the difference in goal representation.
2. Would it be possible to train Phase 1 and Phase 2 together or in an end-to-end fashion? This would provide a ‘coupled’ version that is different from any of the baselines studied in the paper because it still uses the semantic configuration as the intermediate goal representation while having joint training of the language representation and the sensorimotor. If this baseline struggles to learn (possibly due to difficult optimization/local minimas), then this will help further strengthen the thesis of the importance of decoupling the learning process into two distinct phases.
3. Section 3.2: the main text and appendix C.2 was not very clear about the second inductive bias for the symmetry of the behavior required to achieve $above(o_i, o_j)$ and $above(o_j, o_i)$. Are you saying, for example, if we are trying to have object 1 above object 2, then we specify the goal in the form $g_1$, while if we want object 2 above object 1, then we specify the goal in the form $g_2$?

**Minor comments**:
* When using double quotes in latex, use backticks for the opening quote.

**After rebuttal responses**:

I have read the authors’ updated draft and response to my concerns, as well as the other reviews. The updated paper provides a clearer framing and some missing baselines have also been included. I raised my evaluation to a weak acceptance for the paper.

---

> ### Author Response · Authors · 2020-11-19
> **Answer to Reviewer 4**
>
> We thank Reviewer 4 for their helpful feedback.  Here we answer comments that were specific to Reviewer 4. Concerns shared with other reviewers are addressed in the general answer.
>
> **Concerns about the baselines**
>
> We are indeed using HER in the design of our baselines. The baselines are designed so as to reduce to its minimum the number of potential confounding factors. For this reason, we keep most modules strictly equivalent (HER included). The main answer provides further details about the baselines, including the new Language baseline that will replace the older one.
>
> The reviewer is also asking whether we could train the DECSTR agent with Phase 1 and Phase 2 coupled together or in an end-to-end fashion. As long as there is a fixed intermediate representation, the gradients cannot flow backward, which prevents any end-to-end learning. The new end-to-end language-conditioned baseline implements a state-of-the-art language-conditioned RL architecture and performs this coupled learning. Furthermore, the introduction of the language-conditioned goal generator makes the emergence of behavioral diversity possible, as many valid configurations can be generated for a given language input. Other implementations of our architecture could mix the two phases: either run them asynchronously, or make repeated cycles between them, etc. This could allow the tutor to guide its selection of instructions so as to orient the sensorimotor learning of the agent.
>
> **Organization of the method section**
>
> As explained in the main answer, we deeply reorganized the method and experimental sections. This was made easier by the reflection on the focus of the paper, as outlined in the main answer. Important details are also moved back from the appendix to the main document to facilitate comprehension.
> We fixed backticks, thank you.
>
> **Conclusion**
>
> It seems Reviewer 4 is mostly concerned about the design of the baselines. Answering their concern, we declared using HER in our baselines. The main answer details improvements made on both baselines: improving the position baseline and redefining the language baseline to better support our main contribution. We hope the new organizations of the method and experiment section help understanding.

---

> > ### Comment · AnonReviewer4 · 2020-11-25
> > **Thank you for the clarifications and update. Some additional comments and questions.**
> >
> > Thank you for your response and the updated draft. The use of HER in the baselines is much clearer now, and I appreciate the new language conditioned RL (LC-RL for L->B) baseline in Section 4.2 which answers the question about the need for intermediate baselines, and the LGB-C baseline which answers the question about the need for semantic intermediate representation compared to continuous representation.
> >
> > Note here that the continuous goal representation in LGB-C is the (randomly sampled) specific 3D coordinates of the blocks that satisfy the semantic configuration (i.e. it is one of many possible specific configurations that satisfies the semantic configuration). For the G->B phase, my intuition here is that since the specific configuration is sampled randomly for the semantic configuration, it makes sense that the LGB-C version ends up moving the blocks more / takes longer since we are moving it to any one of the valid specific configurations, rather than inferring the ‘closest’ valid configuration. Am I understanding this correctly?
> >
> > On the L->G part, do you have some more intuition on why there is low diversity/recall? I wonder if it has to do with the capacity of C-VAE used in LGG? I am presuming that the interaction data used to train LGG C-VAE has enough diversity (many configurations for the semantic), but sampling from C-VAE has low diversity in the continuous goal configuration here. I am curious to see what the authors think.
> >
> > Hopefully, there is enough time for the authors to respond. Given the major changes which improved the paper clarity and methodology, I am willing to consider increasing my score to a weak acceptance (still contemplating).

---

> > > ### Author Response · Authors · 2020-11-25
> > > **Answer to Reviewer 4**
> > >
> > >
> > > Thank you for this quick answer.
> > >
> > > **About LGB-C**
> > >
> > > LGB-C indeed samples random targets, which forces a minimal distance for blocks to travel. This is the way it is implemented in Li et al., 2019 and Lanier et al., 2019, probably because it increases diversity in training trajectories. We agree that we could in principle add another module to sample closer targets at test time. DECSTR however, does it spontaneously as it learns to reach semantic goals and does not require an extra module. We are rerunning the analysis with the extra module and will report the results in the camera-ready version (and here if time allows).
> > >
> > >
> > >
> > > **About LGG module for continuous targets**
> > >
> > > It is true that the dataset contains a good diversity of continuous targets. We use the same dataset to generate semantic configurations or continuous targets, but it is much more diverse for the continuous targets. Indeed, considering semantic configurations leads to many repeats in the dataset, while continuous targets are always new.
> > >
> > > It seems the C-VAE has trouble integrating the language condition to differentiate between different target distributions and mostly predicts a low-diversity of average targets resulting in a semantic configuration where all blocks are close together. We did try to add more capacity to the network (up to three layers of 256), and investigated other loss functions (soft BCE or the continuous Bernoulli from https://arxiv.org/abs/1907.06845 , both with normalized targets in [0, 1]), but this did not help. Further investigations of this issue might help improve the language-conditioned goal generation. However, leveraging internal semantic representations will always be easier, as there is a more direct mapping between language and this semantic representation. It might seem ad-hoc at first, but language evolved as a way to express these internal semantic representations to other humans. Furthermore, the discrete aspect of semantic representation helps generalize to logical combination, which cannot be done directly with continuous goal generation.

---

### Official Review · AnonReviewer1 · 2020-10-29
**The research question and the main contributions are not clear.**

**Rating:** 4
**Confidence:** 3

**Review:**

This paper introduces DECSTR, which is an agent having a high-level representation of spatial relations between objects. DECSTR is a learning architecture that discovers and masters all reachable configurations from a set of relational spatial primitives. They demonstrated the characteristics in a proof-of-concept setup.

In the introduction, the inspiration obtained from developmental psychology is described. Motivation and background are broadly introduced. A wide range of related works are introduced in section 2.
The motivation and target of this paper are ambitious and important.

However, from the "methods" part, i.e., section 3, this paper is hard to follow.
The supplementary material helps to understand. However, I believe some of the informative and detailed information in the supplementary material should come to the main manuscript.

The proposed method, i.e., DECSTR, is comprised of many components. Therefore, the main contribution is also not clear. # What is the main argument of the paper?
Experimental conditions are also hard to follow.

In evaluation, Figure 1 shows ablation studies alone, i.e., comparison with the variants of DECSTR.
Therefore, the contribution of the paper is hard to grasp.

We can understand what kind of task is achieved in this paper.
Currently, the paper somehow seems to be a demonstration of DECSTR.
In this sense, if the authors state research questions, challenges, and contributions of this paper more clearly, that will make this paper more impactful.

---

> ### Author Response · Authors · 2020-11-19
> **Answer to Reviewer 1**
>
> We thank Reviewer 1 for their helpful feedback. Here, we answer comments that were specific to Reviewer 1. Concerns shared with other reviewers are addressed in the general answer.
>
> **The method section is hard to follow**
>
> As detailed in the main answer, the new organization of the method section presents the general LGB architecture, the environment, then follows the three modules of the proposed instantiation of the LGB architecture with DECSTR:
> * the semantic representation;
> * the intrinsically-motivated goal-conditioned RL algorithm
> * the language-conditioned goal generator.
>
> This new organization should be easier to follow. In addition, we moved part of the information from the appendix back to the main paper to facilitate comprehension.
>
> **Ablations and baselines**
>
> We call “baselines” variants of our algorithm that implement defining features from related state-of-the-art approaches (language-conditioned RL and continuous goal conditioned RL for manipulation tasks respectively). We call “ablations” variants of DECSTR that aim at showing the importance of its components (inductive biases, curriculum, etc.). Our main answer and the new version of the paper clarify the purpose and definitions of our two baselines.
>
> **The paper seems to be a demonstration of DECSTR**
>
> We agree that the focus of the paper was not clear in the previous version. We hope the new positioning outlined in the main answer helps resolving these concerns. The paper will be about the novel RL architecture LGB. Most of its properties emerge from its design (decoupling, goal generation leading to behavioral diversity and enabling strategy switching). DECSTR is thus a concrete illustration of these properties in a specific setup. We argue that other implementations following the same overall architecture will demonstrate similar properties and benefits over existing approaches.
>
> **Conclusion**
>
> It seems the main concerns of Reviewer 1 are about the contribution and challenge statements as well as the organization of the methods and experimental section. We hope that our answers and the new version of the paper help resolve them.

---

### Author Response · Authors · 2020-11-19
**General answer to all reviewers (1/3)**

*We answer here the comments shared by several reviewers. Additional comments specific to each reviewer are answered in specific answers.*

We sincerely thank all reviewers for their very useful feedback.

Most reviewers acknowledged the relevance of our topic of interest (R1, R2, R4), found the approach well motivated (R1, R2, R4) with a strong related work section (R1, R4). R2 found the paper well written (although not well organized) and R4 acknowledged the quality of our experimental section that presents all ablations and a detailed study of generalization properties.
However, all the reviewers also agreed that the paper lacked a clear statement describing its main focus or contribution. In turn, this led to a poor organization of the method and experiment sections. We agree with all these comments and deeply reorganized the paper to answer these concerns. This answer defines the problem we aim to tackle, clearly states our contributions towards its resolution, and describes the reorganization of the methods and experimental sections to support our claims.

**The problem (all)**

Our main goal is to design agents that can learn both on their own and under the guidance of a (human) tutor. To learn on their own, these agents need to generate and pursue their own goals, and to learn from their own reward signals. To learn under the guidance of a tutor, they need to learn to fulfill language-based instructions after interacting with a tutor.

Most current approaches cannot generate goals themselves, and require externally-provided rewards. Language-conditioned RL approaches, especially, almost always require external instructions and rewards. An exception is IMAGINE (Colas et al., 2020) which combines intrinsically motivated language goal generation and internal rewards.

Nevertheless, the direct conditioning of the policy on language inputs in language-conditioned RL approaches (IMAGINE included) imposes some limitations:
1. the agent cannot learn to behave before it starts acquiring language,
2. direct conditioning leads to low behavioral diversity for a given language input,
3. a direct consequence of 2) is that the agent cannot switch strategy for a given instruction.

By contrast: 1) pre-verbal infants demonstrate goal-directed behaviors (Mandler, 2012); 2) humans can find a diversity of ways to fulfill an instruction; 3) they can switch strategies if the first strategy failed.

**Our contributions (all)**

**The main contribution** of this paper is to present a new intrinsically motivated RL architecture called Langage-Goal-Behavior (LGB). LGB tackles the problem above and demonstrates the three properties. It differs from standard language-conditioned RL by the introduction of an intermediate semantic goal representation (G) between language inputs (L) and behavior (B). This intermediate representation allows the decoupling of language and behavior. Agents can either learn autonomously to target semantic configurations **or** learn to follow instructions by mapping language-based instructions to their semantic goal representation space. We argue LGB demonstrates the 3 properties.

**Our second contribution** is the DECSTR learning algorithm: a particular instance of the LGB architecture for manipulation domains. The paper argues that the 3 properties above emerge from the LGB architecture, given sufficiently efficient components: 1) a semantic representation that characterizes interesting behaviors; 2) an intrinsically motivated goal-conditioned RL algorithm that can learn to reach semantic configurations and 3) a language-conditioned goal generator with good precision and recall. DECSTR illustrates that, when these conditions are met, the three properties emerge in the system. We do not claim that DECSTR is the most efficient instance of LGB, and future LGB implementations may benefit from improvements in the fields of goal-conditioned RL and/or generative modelling.

Finally, some technical aspects of the DECSTR implementation can be seen as additional but minor contributions: the novel curriculum learning strategy and the language-conditioned goal generation module based on C-VAE.

We reframed the paper along these lines. This clearly states the problem targeted by our system (answering R2), what our contributions are (answering R1 and R3) and better focuses the paper.

---

### Author Response · Authors · 2020-11-19
**General answer to all reviewers (2/3)**

**Paper organization (all)**

The method and experimental sections are now reorganized to reflect the new positioning of the paper and information facilitating comprehension are moved back from the appendix to the main document. The new organization is this one:
* The method section presents the general LGB architecture, the environment and the implementation of the three modules composing LGB architecture in the DECSTR algorithm: 1) the semantic representation; 2) the intrinsically motivated goal-conditioned RL algorithm and 3) the language-conditioned goal generator.
* The experimental section is reorganized in three sections (R1,R2,R3,R4):
   * S1 shows that DECSTR solves the task
   * S2 compares DECSTR to a language-conditioned RL approach and show the three properties emerge only in DECSTR
   * S3 shows that LGB benefits from semantic representation when compared to a variant using continuous representations.


**Baselines (R1, R2, R4)**

Our problem requires agents to learn from two separate sources (language instructions and self-generated goals). This limits the use of existing algorithms as-is. In the design of our two baselines, we decided to integrate the defining features of state-of-the-art algorithms in variants of ours (see precisions below). This strategy has two benefits: 1) it helps control for confounding factors that can emerge from the use of a completely different architecture and code base; 2) it mitigates the problem of under-tuned baselines, as most components are shared across algorithms. Here we clarify the purpose of our baselines:
* The language-conditioned baseline (LB) is used to compare the LGB architecture (implemented as DECSTR) to standard language-conditioned algorithms, focusing on their performance on the three properties listed above. By comparing the two approaches on the same instruction-following task (turning 102 instructions into corresponding behaviors), we study the properties emerging from the decoupled LGB architecture. We are currently implementing this new version of the baseline. We will report the results and update the paper as soon as possible. We hope to show that, even if it succeeds in the instruction following task, it will not show behavioral diversity nor strategy switching behaviors, compared to DECSTR.
Note that this baseline replaces the previous Language baseline, whose purpose was unclear. The new baseline implements defining features of IMAGINE (Colas et al., 2020), without goal imagination, but with an oracle reward function and can thus be seen as a state-of-the-art intrinsically motivated language-conditioned algorithm.
* The Position baseline. The purpose of this baseline is to demonstrate the benefits of decoupling via a semantic representation instead of a continuous one. In addition to the non-binary suggested by R2 (that did not help alone), we added another defining feature from Lanier et al., 2019 (multi-criteria HER) and an additional object-centered inductive bias, making this baseline closer to state-of-the-art goal-conditioned RL algorithms for block manipulation. We thank R2 for pushing us to spend more time on this baseline. The sensorimotor learning phase now demonstrates good results. The LGB architecture still seems to benefit from semantic representation: 1) for interpretability (it is more natural to ask “put the red block on the blue block” than asking “put the block 1 at (1.2, 0.9, 2.3)”); 2) for language acquisition 3) to facilitate opportunistic goal completion and 4) to acquire skill repertoires. Indeed the behavior of this baseline can be seen as a unique skill: placing blocks at their targets. This does not discriminate between different semantic skills.

---

### Author Response · Authors · 2020-11-19
**General answer to all reviewers (3/3)**

**On the definition of semantic representations**

In this new positioning, the set of semantic predicates is not a contribution. We do not argue that our proposed representation is general enough to solve all tasks.

Extending the set of spatial predicates handled by an LGB architecture is an interesting question for future research. Indeed, infants spend most of their time in what can be considered as manipulation scenarios. We know from Mandler 2012 that infants use spatial predicates really early in life, and she even argues that a small set of them (around 20) enables infants to bootstrap an important set of sensorimotor and cognitive skills.

On the application of LGB architectures to other domains, we argue that the definition of sets of semantic predicates (i.e. binary sensors) is easier and involves less prior knowledge than the definition of goal spaces and associated reward functions it replaces. Indeed, defining the semantic predicates is defining the dimensions of a behavioral space. It does not require the engineer to fully grasp all behaviors in that space, to know which behavior can be achieved and which ones cannot, nor to define reward functions for each of them. The space of potential behaviors becomes combinatorially larger as new semantic predicates are added, the reward function only asserts equality between the current and goal configurations. Agents could also easily grow the space of potential behaviors by adding new semantic predicates across learning.

**Conclusion**

It seems the main concerns of the reviewers were about the lack of clear problem and contribution statements and a poor organization of the methods and experiments sections. Thanks to their comments, we believe the new positioning of the paper and its new organization that results from it answer these concerns. We hope reviewers will find the time to read the updated version of the paper, that includes all the points discussed above and presents a clearer organization.

---

> ### Comment · AnonReviewer3 · 2020-11-22
> **Revised paper incomplete**
>
> Will the complete version of the revised paper be ready before the deadline for discussion? The new framing seems promising, but the paper still has "coming soon" notes and a lot of stray-looking paragraph headings breaking up the text.

---

> > ### Author Response · Authors · 2020-11-23
> > **Revised version nearly complete**
> >
> > The revised paper is nearly complete. After taking into consideration the reviewers' comments, we updated ~80% of the previous version.
> >
> > We will have the last results of the Language baseline by the end of tomorrow and will update the two corresponding paragraphs (in Section 4.2) accordingly. Given the current progress of the runs, we are expecting it to learn all the close and far goals but not to succeed in the stacking goals. In the mastered goals, we are also expecting a reduced diversity compared to DECSTR.
> >
> > The rest of the paper is definitive and will not be updated before tomorrow's deadline.

---

### Author Response · Authors · 2020-11-24
**Final rebuttal revision**

One more time, we thank the reviewers for their efforts and constructive feedback.

We integrated the last results of the language-conditioned baseline to the paper.

The reviews helped us improve the positioning of the paper. The new version now clearly presents the targeted problem and states our contributions towards its resolution. We undertook a major rewriting of the paper to better convey these ideas, ran complementary experiments (the language-conditioned baseline) and improved our position baseline.

We hope the reviewers will find time to go through the revised version.

---

### Decision · Program_Chairs · 2021-01-07
**Final Decision**

**Decision:**

Accept (Poster)

**Comment:**

This paper presents a new approach to grounding language-based RL tasks via an intermediate semantic representation, in an architecture called language-goal-behavior (LGB).  The architecture permits learning a mapping from internal goals to behavior  (GB) separately from learning a mapping from language to internal goals (LG), and prior to flexibly combining all three (LGB).  The architecture is studied in a specific implementation called DECSTR.  The architecture has multiple desired attributes including support for intrinsic motivation, decoupling skill acquisition from language grounding, and strategy switching.  The experiments demonstrate the utility of different components in the architecture with a variety of ablation results.

The reviews initially found the paper to be poorly organized with required content described only in the appendix (R1, R2, R4), with unclear main contributions (R1, R2, R4), and with results restricted to demonstrations (R3).  Despite these reservations, the reviewers found the content to be potentially relevant though narrow in scope.

The authors substantially revised the paper. They improved its organization, clarified contributions, separated the architecture from the specific examples, and improved the experimental baselines.  After reading the revised paper, the reviewers agreed that the paper's organization and insights were improved, making the new paper's contribution and insight clear.  The experimental baselines were also improved, providing more support for the potential utility of the proposed method.

Three reviewers indicate to accept this paper for its contribution of a novel approach to grounding language and behavior with an intermediate semantic representation. No substantial concerns were raised on the content of the revised paper. The paper is therefore accepted.